# A Real-Time Measurement Method of Air Refractive Index Based on Special Material Etalon

**Guo-Ying Ren [1,2,\*], Xing-Hua Qu [1] and Shuang Ding [1]**

[1] State Key Laboratory of Precision Measurement Technology and Instruments, Tianjin University, Tianjin 300072, China; quxinghua@tju.edu.cn (X.-H.Q.); dingshuang0626@163.com (S.D.)
[2] National Institute of Metrology, Beijing 100029, China
\* Correspondence: rengy@nim.ac.cn

**Abstract:** In the precise displacement measurement based on laser interferometry, the measurement technology for the refractive index of air is widely used to improve the measurement accuracy. However, the existing measurement method of the refractive index of air based on direct measurement is not easy to realize in practical work because of its complex measurement principle and the huge volume of the measurement device; while the measurement accuracy and speed based on the indirect method cannot adapt to the real-time, fast and accurate measurement requirements of industrially changing environments, resulting in distortion of the results. In this study, a measurement method of the refractive index of air based on a special material etalon is proposed. The method enables rapid and direct measurement of the air refractive index when the environment changes and it is given the realization process. Finally, the experimental results show that the deviation between this method and the modified Edlen formula is about $2.5 \times 10^{-7}$, and that this method can quickly reflect the changes of the environment, which prove the correctness of this method and its ability manage rapid environmental responses. This method is worth popularizing in industrial measurement.

**Keywords:** metrology; precision measurement; air refractive index; wavelength correction

## 1. Introduction

During the displacement measurement based on the principle of laser interferometry, the refractive index of air in the optical path is often changed because of the change of ambient atmosphere, but it is difficult to measure the refractive index of air rapidly and accurately because of the limitation of the conditions or measurement schemes [1]. According to the principle of laser interference:

$$L = \frac{K \lambda_0}{2n} \tag{1}$$

where $L$ is the length of the measurement, $K$ is the interference level, $n$ is the refractive index of air, and $\lambda_0$ is the vacuum wavelength.

It is known that the variation ($\Delta L$) of the displacement measurement $L$, caused by the variation ($\Delta n$) of air refractive index $n$, is as follows:

$$\Delta L = \frac{K \lambda_0}{2 n^2} \Delta n \tag{2}$$

Therefore, for precise measurement, the change in the air refractive index $n$, caused by the change in environmental factors and the inaccuracy of its measurement, become one of the main reasons for the limitation on the improvement of the accuracy of the measurement results. How to measure the

air refractive index accurately and quickly in order to compensate for the limitation is the primary consideration in the scheme design of measurement.

At present, there are two ways to obtain a measurement of the air refractive index: The direct method and the indirect method. The direct method is to calculate the air refractive index by directly comparing the optical path difference between the air cavity and the same-length vacuum cavity with the refractive index measuring instrument or laser interference technology [2,3], which can reach a $10^{-8}$ level of measurement accuracy. However, its design and fabrication is complex and the vacuum required by the experiment is very high. At the same time, the atmospheric pressure makes the vacuum system easy to deform and re-introduce the error source, so it is seldom used in engineering. The indirect method is to measure the parameters of the air composition through various sensors, such as ambient temperature, humidity, air pressure and so on. The refractive index value is calculated in order to obtain the air refractive index formula [4–12]. The important characteristic of the indirect measurement method is that it is convenient and easy to use. The measurement accuracy is $10^{-7}$ level but because of the delay of the sensor itself, the air refractive index value calculated after the measurement is not the true value at that time. The measurement system accuracy is also low and is prone to produce errors in fact. Some errors are up to a $10^{-6}$ level in some measurement systems.

In order to synthesize the advantages of the above two methods and avoid their disadvantages, this paper presents a new measurement method for the refractive index of air based on a special material etalon.

## 2. Theory

### 2.1. Principle of Measurement

From Equation (2), it can be derived that:

$$\Delta n = n_1 - n_0 = \Delta L \frac{n_0}{L} \tag{3}$$

where $n_0$ is the initial measurement value of the air refractive index and $n_1$ is the current measurement value of the air refractive index.

So, the current value $n_1$ of refractive index of air can be calculated as:

$$n_1 = n_0 + \Delta n = n_0 + \Delta L \frac{n_0}{L} = n_0 \cdot \left(1 + \frac{\Delta L}{L}\right) \tag{4}$$

According to this principle, a type of zero-expansion quartz material with the known coefficient of thermal expansion and its value close to zero, was selected to design and manufacture a standard etalon work-piece (as shown in Figure 1) with length $L$. It was placed in the measuring light path in order to monitor the variation of the wavelength of the light path caused by the fluctuation of the environment during the measurement.

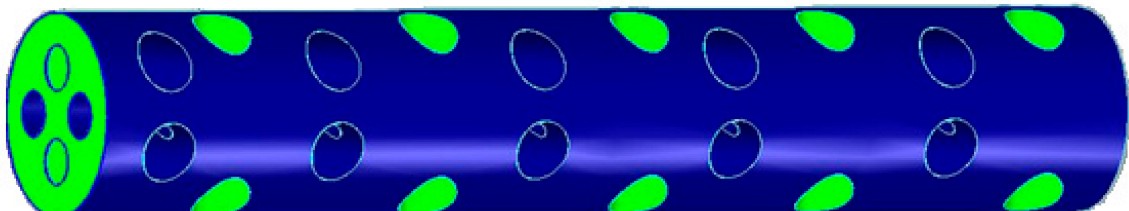

**Figure 1.** Standard etalon work-piece for measuring the refractive index of air.

The front-end face of the standard etalon was coated with reflective film as a reference arm, and the back end of the standard etalon was coated with reflective film as a measuring arm. The layout of the reference light path and the measured light path based on the etalon are shown in Figure 2 [13].

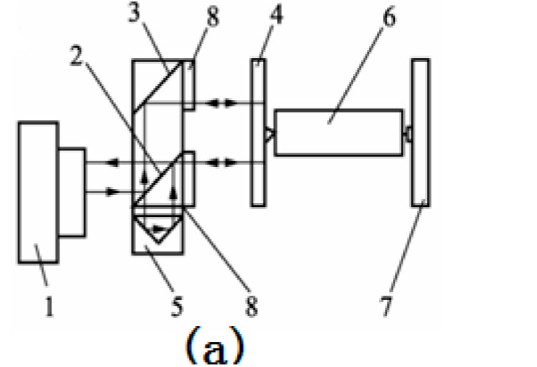 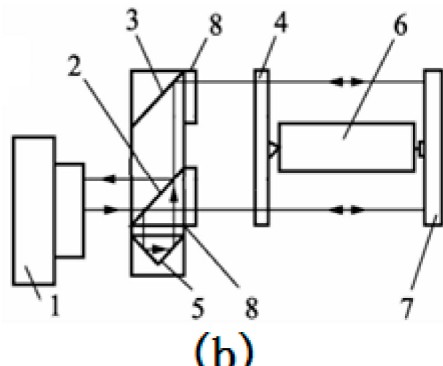

**Figure 2.** The layout of light path based on the etalon. (**a**) Description of the layout of the reference light path based on the etalon. (**b**) Description of the layout of the measurement light path based on the etalon.

In Figure 2a, the beam is emitted from laser one, reflected from the reflector mirror two to beam bender mirror three, passed through the quarter wave plate eight, which is incident to measuring mirror four fixed on the front face of the tested piece six, returned from four and passed through two, reflected from two to four again through cube corner mirror five, and returned from four to form a reference beam through two. The measurement optical path in Figure 2b can be analyzed similarly.

As shown in Figure 2, the measured beam and the reference beam pass through the same optical path in the interferometer. They are designed based on the concept of the common light path and symmetrical distribution of the interferometer. So, the measurement system has a heat balance function, that is to say, when the size of the interferometer varies with the temperature effect, the influence on the length measurements of the two light paths are cancelled out one another.

### 2.2. Measurement Model

In order to obtain an accurate initial $n_0$, the indirect measurement method of the air refractive index should be used first. Therefore, some high-sensitivity temperature, pressure and humidity sensors, and $CO_2$ water vapor sensors were arranged in the measuring optical path in order to measure the air temperature, pressure and humidity, and then the initial refractive index $n_0$ was calculated and obtained by the refractive index correction formula.

Because the conventional Edlen formula for calculating the air refractive index does not include the modified item of the $CO_2$ error [14,15], the improved Edlen formula is as follows:

$$(n-1)_s = 10^{-8} \cdot [A + B/(130 - \sigma^2) + C/(38.9 - \sigma^2)] \tag{5}$$

$$(n-1)_x = [1 + 0.540 \cdot (x - 0.0003)] \cdot (n-1)_s \tag{6}$$

$$n_{TP} = 1 + P \cdot (n-1)_x / D \cdot [1 + 10^{-8} \cdot (E - F \cdot T) \cdot P] / (1 + G \cdot T) \tag{7}$$

$$Y = -J \cdot [1 - (T/273.16)^{-1.5}] + K \cdot [1 - (T/273.16)^{-1.25}] \tag{8}$$

$$n_{Tpf} - n_{Tp} = -(f/100 \cdot 611.657 \cdot e^Y) \cdot (292.75/T) \cdot [3.7345 - 0.0401 \cdot \sigma^2] \cdot 10^{-10} \tag{9}$$

Here, the constants can be defined as follows [11]:

$A$ = 8342.54; $B$ = 2,406,147; $C$ = 15,998; $D$ = 96,095.43; $E$ = 0.601; $F$ = 0.00972; $G$ = 0.003661; $J$ = 13.928169; $K$ = 34.7078238; $\sigma = 1/\lambda$; $T = t + 273.15$.

Here $(n-1)_s$ is the refractive index of air in the standard state, $\sigma$ is the wave number in the vacuum, expressed in the unit of $um^{-1}$, $(n-1)_x$ is the refractive index of air with the molar content of carbon dioxide, where the unit is $(\times 10^{-6})$, $n_{Tp}$ is the refractive index of air under the conditions of temperature, air pressure and carbon dioxide, and $n_{Tpf}$ is the refractive index of air under the conditions of temperature, humidity, air pressure and carbon dioxide.

After obtaining the initial value of the air refractive index, the etalon can be utilized to calculate the real-time wavelength compensation. The compensation factor $C$ is:

$$C = \frac{10^6}{n_{Tpf} + 10^6} \tag{10}$$

Suppose that the number $P_n$ of preset wavelengths at the moment of measurement is:

$$P_n = \frac{LRf}{C\lambda} \tag{11}$$

where $L$ is the effective length between the two mirrors of the etalon of the air refractive index, $R$ is the electrical resolution of the interferometer, and f is the fold factor of the laser interference in the light path.

The value of the compensation $C_n$ is therefore:

$$C_n = \frac{LRf}{(P_A + P_n)\lambda} \tag{12}$$

where $P_A$ is cumulative count.

The dead-path count $C_{Dn}$ is:

$$C_{Dn} = \frac{DRf}{C\lambda} \tag{13}$$

where $D$ is the dead-path length in the interferometer optical path.

Then the initial actual displacement $P_{s0}$ after compensation is:

$$P_{s0} = \frac{C_n(P_A + C_{Dn})\lambda}{Rf} \tag{14}$$

Finally, the current value $n_1$ of the reflective index of air is:

$$n_1 = \left(1 + \frac{\Delta L}{P_{s0}}\right) \tag{15}$$

## 3. Experiments and Results

In order to verify the correctness of the above formulae of the air refractive index based on the etalon, the verification scheme of measurement of the optical path of the air refractive index was constructed as follows (see Figure 3).

The light from the laser passes through the beam splitter (BS) mirror and divides into two paths. One path passes through the beam bender mirror and the first interferometer to the air refractive index standard (etalon). The returned light passes through the first interferometer and returns to the electronic signal subdivision circuit (SPC). The processed signal data is collected by the computer. The other light path passes through the second interferometer to the cube corner reflector. It is reflected back to the second interferometer and back to the SPC. The processed signal data is collected by the computer. At the same time, the environmental sensors are arranged on a second path in order to calculate the initial value $n_0$ of the refractive index of the air. In particular, these two light paths can be placed close to each other, so it can be thought that the two light paths have the same environmental conditions and the same refractive index of the air.

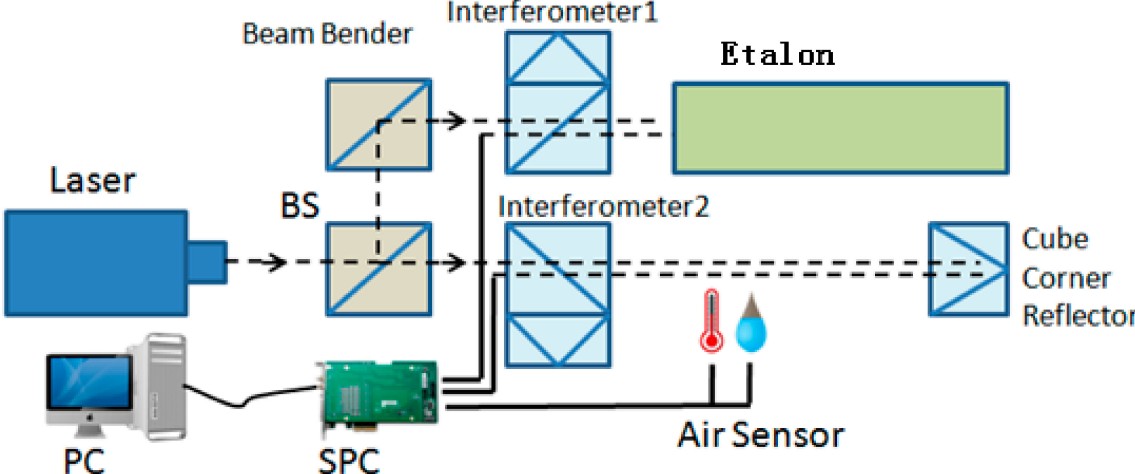

**Figure 3.** The verification scheme for the measurement optical path of the air refractive index. BS: beam splitter, SPC: signal subdivision circuit, PC: personal computer.

In the experiment, a Renishaw RLE10 was selected as the laser source, which has a laser signal with 64 times electronic subdivision. The actual measurement value of the thermal expansion coefficient of the Etalon used in the air refractive index standard tool is $6.1 \times 10^{-8}$/K (see Figure 4). The variation of the thermal expansion coefficient of the Etalon in the range of 20 °C $\pm$ 5 °C is not more than $1.0 \times 10^{-8}$/K, which guarantees the outstanding thermal stability of the length.

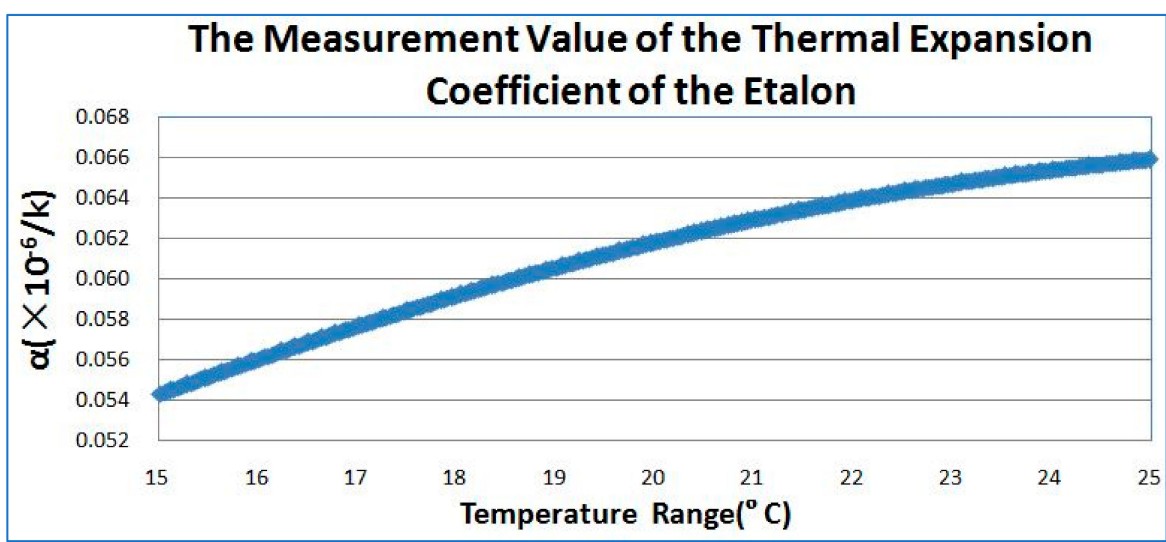

**Figure 4.** The measurement value of the thermal expansion coefficient of the Etalon.

The diagram comparing the variation between the measured values of the air refractive index obtained from the Etalon in the first light path and the calculated value of the air refractive index from the ambient atmosphere sensors in the second path are shown in the following figure (see Figure 5).

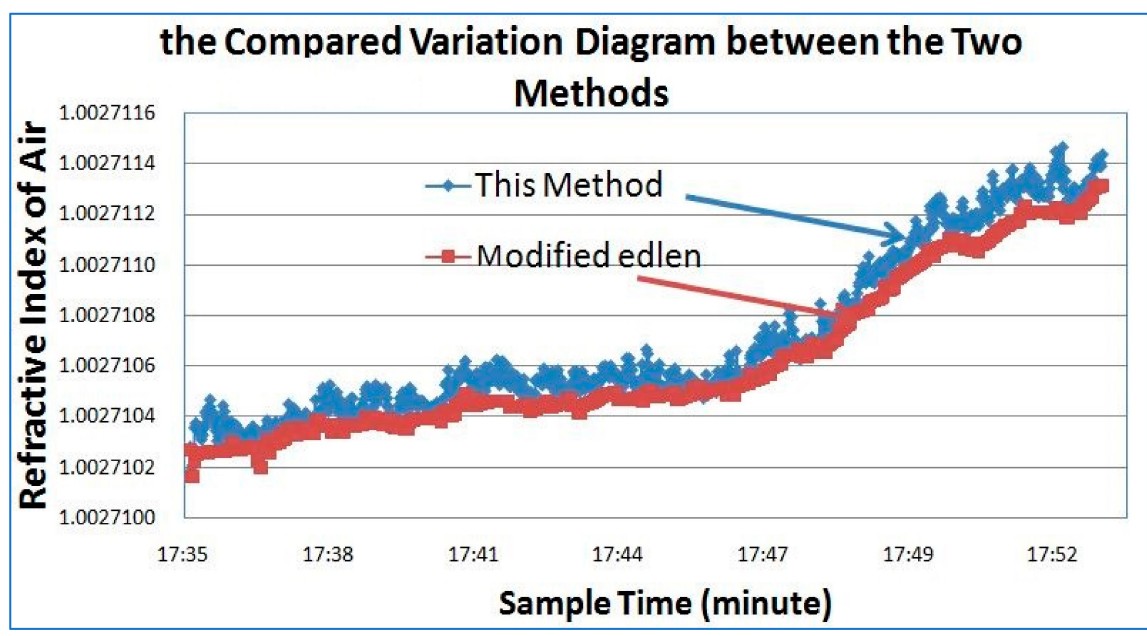

**Figure 5.** The compared variation diagram between the two methods.

It can be seen from Figure 5 that the change in the air refractive index calculated by the modified Edlen formula is relatively gentle because of the hysteresis of the sensors, which cannot fully reflect the effect of air fluctuation on the actual air refractive index. However, the method proposed in this paper can obtain the real-time changes in the air refractive index easily and quickly, and perform the further modification and compensation in the subsequent length measurement. From Figure 6, it can be seen that the maximum difference between the two measurement methods is up to $2.5 \times 10^{-7}$.

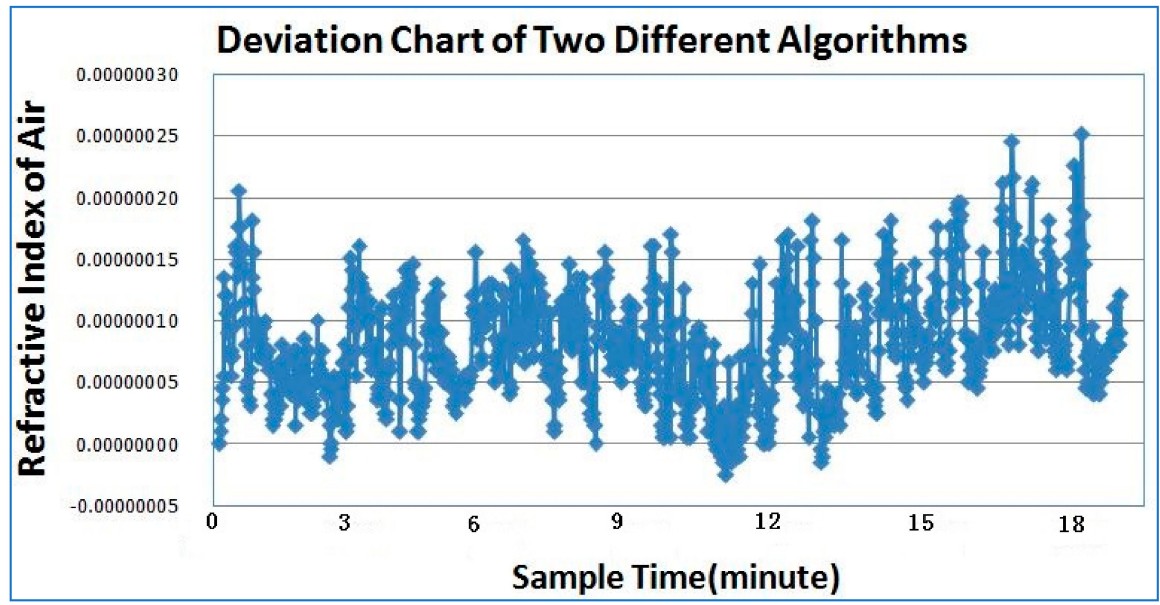

**Figure 6.** The maximum difference between the two measurements.

## 4. Conclusions

From the above analysis it can be seen that the requirement for length measurement accuracy becomes higher and higher in the precision measurement. However, the current measurement methods for the air refractive index based on Edlen formula cannot provide real-time measurement results because of the ambient atmosphere sensors it requires. In this paper, a new method for air refractive

index measurement is proposed, which can rapidly measure a changing air refractive index in real time, and provides real-time wavelength correction data for high-precision length measurement. Finally, the correctness of the method is proven by the experimental results. In future we will attempt to analyze and describe the high-frequency noise problem which might have been present in the described method, although the noise may have been very small.

There is hope that this measurement method can provide a reference for others who need to improve the accuracy of laser interferometry in the precision manufacturing industry.

**Author Contributions:** G.-Y.R. conceived the method, improved the experiments and wrote the paper; X.-H.Q. conducted the experiments; S.D. curated the data.

**Funding:** This research was funded by the National Key Research and Development Program of China, grant number: 2018YFF0212702.

**Acknowledgments:** This study was supported by the National Key Research and Development Program of China (Grant No.: 2018YFF0212702). The authors would like to thank the other members of the research team for their contributions to this study.

**Conflicts of Interest:** The authors declare no conflict of interest.

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
