# Peer review of "A Real-Time Measurement Method of Air Refractive Index Based on Special Material Etalon"

_applsci, doi:10.3390/app8112325_

Round 1
Reviewer 1 Report
This is a great method that you've come up with! I'm quite excited to see someone come up with a method like this to measure n in media, especially given the lower device volume that you've come to. This is excellent work well done; thank you for your contribution.
Please make certain to go over your grammar with a bit more of an eye for detail, and also check that you're only using the third person (so no "I", "us", "we", etc.).
Author Response
Response 1:
Thank you for your support, recognition and encouragement. I have checked and optimized the paper and grammar again, and determined to use the third person only.
All the modified sections are as listed below.
In 61st row, “Derived from the above formula (2), we can see that” is modified as” Derived from the above formula (2), it can see that:”
In 90th row, “In order to obtain accurate initial n0, we use the indirect measurement method of air refractive index” is modified as “In order to obtain accurate initial n0, the indirect measurement method of air refractive index should be used first”.
In 108th row, “After got the initial value of the air refractive index, we can utilize the etalon to calculate the real-time wavelength compensation” is modified as “After got the initial value of the air refractive index, the etalon can be utilized to calculate the real-time wavelength compensation”
In 140th row, “so we can think that the two light paths have the same environmental conditions and the same reflective index of the air” is modified as “so it can be thought that the two light paths have the same environmental conditions and the same reflective index of the air”.
In 142nd row, “In the experiment, we select Renishaw RLE10 as the laser source and its laser signal has 64 times electronic subdivision” is modified as “In the experiment, Renishaw RLE10 is selected as the laser source and its laser signal has 64 times electronic subdivision”.
In 158th row, “From the fig.6, we can see that the maximum difference between the two measurement methods is up to 2.5 x 10-7” is modified as “From the fig.6, it can be seen that the maximum difference between the two measurement methods is up to 2.5 x 10-7”.
In 163rd row, “From the above analysis, we can get that the requirement of length measurement accuracy is higher and higher in the precision measurement” is modified as “From the above analysis, it can be got that the requirement of length measurement accuracy is higher and higher in the precision measurement”.
Thank you for your support again.

Reviewer 2 Report
The paper deals with a subject of interest in the specific sector of precise measurements. The method proposed is interesting despite the description of the methodology can be further improved. For instance in Fig.2 not all the points are described.
In Eq.1 please, specify better what is intended for "Interference level"
Author Response
Response 1: Thank you for your support and recognition, especially for your very good review opinions.
From fig.2, we can see that it is a measurement laser light path diagram, and symbol 1 represents 633nm wavelength laser, symbol 2 represents the reflector mirror, and symbol 3 represents beam bender mirror, symbol 4 represents measuring mirror, symbol 5 represents cube corner mirror, symbol 5 represents the tested workpiece, symbol 7 represents reflector mirror, symbol 8 represents 1/4 wave plate.
In Eq.1, K is the interference level, that is to say, when measuring displacement with laser interferometry principle and satisfying the condition of laser interferometry, K is the number of alternating light and shade changes in the interference fringe pattern formed by laser reference light path and laser measurement light path.

Reviewer 3 Report
Change in use of mathemetical operator representation: eq 1-4 vs eq 5 ff
Especially use of x as multiplictor symbol is misleading
Layout of eq. 5-8 should be improved, really hard to read
Graphical representation Fig 4 and 5 should be improved: - same font size for both, transparent marker symbols, axis-labels clearly, more descriptive caption
Conclusion "...which can measure rapidly changing air refractive index in real time..." is ambitious, since presented measurement results only show a high frequency component. This might be result of real changes in refractive index, but other sources like noise are likely as well.
Author Response
Response 1: Thank you for your support and your five good review opinions.
All the questions you mentioned have been answered as follows.
For question 1, all places of x as multiplictor symbol have been replaced by *symbol.
For question 2, the equations 5-8 have been improved. The modified version is as the attached file.
For question 3, Fig 4 , 5 and 6 have be improved as the same font size, the same font style, and clearly as possible as it can, and the more descriptive caption.
For question 4, the respectable reviewer thinks that the presented measurement in this paper might be the sources like noise. I want to give some information based on the analysis of measurement uncertainty of the two different methods to discuss it.
This below table(see the attached file) is the uncertainty source of this method of this paper when the measured length of Etalon is 200mm.
From this uncertainty source list of measurement, we can know that the maximum error source is from laser wavelength accuracy and its stability, and the electronics noise. And the expanded uncertainty is 11.4nm, k=2. That is to say, the measurement uncertainty of this method is less than the deviation of the two measurement methods mentioned in the paper.
So, we can think that the method proposed in this paper can get the real changes in refractive index.
But, it is very important for me that how to separate the high frequency component of the result mentioned in your reviews. I will do more work for it in the future.
So, the last paragraph is be modified as follows.
“From the above analysis, it can be got that the requirement of length measurement accuracy is higher and higher in the precision measurement, but the current measurement methods of air refractive index based on Edlen formula cannot provide real-time measurement results because of its used ambient atmosphere sensors. In this paper, a new method of air refractive index is proposed, which can measure rapidly changing air refractive index in real time, and provides real-time wavelength correction data for high precision length measurement. Finally, the correctness of the method is proved by the experimental results. Next, I will try to analyze and find out the high-frequency noise problem contained probably in this method, although the noise may be very small.
In general, I hope that this measurement method can provide a reference for the other colleagues who need to improve the accuracy of laser interferometry in precision manufacturing industry.”
Thank you for your patient and good reviews again.

Round 2
Reviewer 3 Report
Thanks for the revised version.
A few further comments to improve your paper:
eq. 10 - 14 same issue with operator symbol x vs* like eq 3-9
eq.# 9 exists 2 times (line 102 and 113)
Fig 4 and 5 I propose to transfer headline text to caption
Fig5 timescale hh:mm:ss? I propose to use relative timing in s or min to clearly indicate the scale
Perhaps it would be helpful to also add a zoom in to a smaller portion of time to increase visability. Doing so, a better idea of thermal behaviour vs noise effects could be transferred.
Author Response
Response 1: Thank you very much for your good review opinions again.
All the questions you mentioned have been answered as follows.
For question 1, all places of x as multiplictor symbol in eq. 10-14 have been replaced by *symbol.
For question 2, all the equations behind line 113 in the paper have been renumbered from 10-15.
For question 3, Fig 4 and 5 have be modified according to your good review.
For question 4, the timescale in Fig 5 and 6 has been modified in minute according to your good advice.
The modified sections from question 1 to question 4 are been seen in the revised manuscript.
For question 4, it is a good idea for me and I will try to do this work next according to your constructive proposal.
Thank you for your patient and good reviews again.
